# OPTIMAL CONTROL MEETS FLOW MATCHING: A PRINCIPLED ROUTE TO MULTI-SUBJECT FIDELITY

**Base Models**  **Base Models + FOCUS (Ours)**

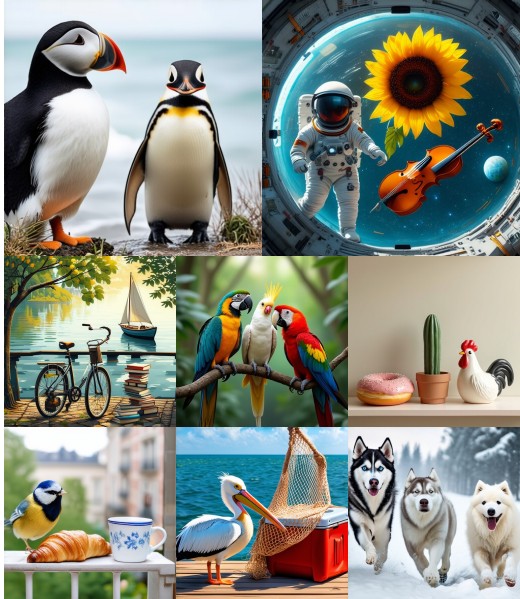

Figure 1: **Optimal control makes flow matching models reliable on multi-subject prompts.** Using FOCUS at test time or via fine-tuning yields faithful multi-subject compositions with correct attributes, minimal leakage, and no omissions, while preserving base style.

## ABSTRACT

Text-to-image (T2I) models excel on single-entity prompts but struggle with multi-subject descriptions, often showing attribute leakage, identity entanglement, and subject omissions. We introduce the first theoretical framework with a principled, optimizable objective for steering sampling dynamics toward multi-subject fidelity. Viewing flow matching (FM) through stochastic optimal control (SOC), we formulate subject disentanglement as control over a trained FM sampler. This yields two architecture-agnostic algorithms: (i) a training-free test-time controller that perturbs the base velocity with a single-pass update, and (ii) Adjoint Matching, a lightweight fine-tuning rule that regresses a control network to a backward adjoint signal while preserving base-model capabilities. The same formulation unifies prior attention heuristics, extends to diffusion models via a flow–diffusion correspondence, and provides the first fine-tuning route explicitly designed for multi-subject fidelity. Empirically, on Stable Diffusion 3.5, FLUX, and Stable Diffusion XL, both algorithms consistently improve multi-subject alignment while maintaining base-model style. Test-time control runs efficiently on commodity GPUs, and fine-tuned controllers trained on limited prompts generalize to unseen ones. We further highlight FOCUS (Flow Optimal Control for Unentangled Subjects), which achieves state-of-the-art multi-subject fidelity across models.

# 1 INTRODUCTION

Text-to-image (T2I) generators have made substantial progress in visual fidelity and prompt adherence, yet they remain brittle on *multi-subject* prompts. Typical failure modes include attribute leakage (an attribute intended for one subject propagates to others), identity entanglement (multiple subjects merged into a hybrid), and subject omission (Chefer et al., 2023; Liu et al., 2022; Bar-Tal et al., 2023; Dahary et al., 2024). These limitations hinder downstream applications such as story illustration, multi-panel composition, and scientific communication, where preserving subject identity and attribute binding is essential.

A unifying theoretical perspective on modern T2I generators is *flow matching* (FM), which parameterizes generation as a time-dependent flow from a base distribution to the data distribution via a learned vector field (Lipman et al., 2023; Liu et al., 2023; Albergo et al., 2023). This framework encompasses both rectified-flow (RF) models used in recent large-scale systems such as Stable Diffusion 3.5 (Esser et al., 2024), FLUX (Labs et al., 2025), and earlier denoising-diffusion architectures such as Stable Diffusion 1.5 (Rombach et al., 2022), Stable Diffusion XL (Podell et al., 2024), enabling statements that transfer across architectures and training choices. We leverage this common ground to analyze—and improve—multi-subject fidelity in FM models.

Prior work has attempted to mitigate entanglement through *test-time* heuristics that reshape cross-attention (Meral et al., 2024) or adjust guidance (Feng et al., 2023), including token amplification (Chefer et al., 2023), constraint-based binding (Li et al., 2023b), and structure-aware attention editing (Hertz et al., 2023; Dahary et al., 2024). While effective in specific settings, these methods are heuristic and lack a unifying optimization objective, making it unclear when and why they succeed. Furthermore, most were developed for Stable Diffusion 1.x backbones, and their portability to RF and modern FM models remains limited.

In this work, we show that multi-subject *disentanglement* can be formulated as a *stochastic optimal control* (SOC) problem for trained FM-based samplers. Concretely, augmenting the base dynamics with a small control that balances proximity to the original generator against a differentiable *disentanglement objective* yields a principled formulation and two complementary algorithms:

(i) **Test-time controller.** A lightweight single-pass controller derived from the optimality conditions of the SOC objective that steers sampling toward disentangled renderings without retraining. The formulation accepts *any* differentiable cost, thereby providing a principled path to adapt existing heuristics to modern FM models.

(ii) **Fine-tuning via Adjoint Matching.** A stable, low-cost update rule based on *Adjoint Matching* (Domingo-Enrich et al., 2025) that regresses a control network onto a backward adjoint signal under a *memoryless* noise schedule, directly minimizing the disentanglement objective while preserving the base model's style and support.

Empirically, our methods improve multi-subject fidelity across both modern FM models (Stable Diffusion 3.5, FLUX) and earlier diffusion backbones (Stable Diffusion XL). The test-time controller provides consistent gains with negligible overhead, while fine-tuning further reduces entanglement without degrading style or generalization beyond the training prompts. Building on these insights, we introduce FOCUS (Flow Optimal Control for Unentangled Subjects), which consolidates our framework into a practical algorithm and achieves the strongest results in our experiments. To foster transparency and reproducibility, we will release code, the curated dataset, and checkpoints of the best-performing fine-tuned models.

# 2 PRELIMINARIES

Flow Matching (FM) (Lipman et al., 2023; Liu et al., 2023; Albergo et al., 2023) trains a time–dependent vector field $v_\theta : \mathbb{R}^d \times [0,1] \to \mathbb{R}^d$ that transports a base distribution $\pi_0$ (e.g., $\mathcal{N}(0, I)$) to a target distribution (e.g. $P_{\text{data}}$), without simulating a forward noising process during training. Given a *reference path* $\overline{\mathbf{X}} = (\overline{X}_t)_{t \in [0,1]}$ with $\overline{X}_0 \sim \pi_0$ and $\overline{X}_1 \sim \pi_1$, FM regresses the *conditional velocity*

$$u_t(\overline{X}_t \mid \overline{X}_0, \overline{X}_1) := \frac{d}{dt}\overline{X}_t \tag{1}$$

so that $v_\theta(x, t)$ matches its conditional expectation $\mathbb{E}[u_t \mid \overline{X}_t = x]$ (Lipman et al., 2023).

**Reference paths.** A standard choice is the linear (Gaussian) interpolant

$$\overline{X}_t = \beta_t \overline{X}_0 + \alpha_t \overline{X}_1, \qquad \alpha_0 = 0, \ \beta_0 = 1, \ \alpha_1 = 1, \ \beta_1 = 0, \tag{2}$$

where $(\alpha_t, \beta_t)_{t \in [0,1]}$ is a differentiable scheduler with $\alpha_t$ strictly increasing, and $\beta_t$ strictly decreasing. The pathwise derivative is then $u_t(\overline{X}_t \mid \overline{X}_0, \overline{X}_1) = \dot{\beta}_t \overline{X}_0 + \dot{\alpha}_t \overline{X}_1$.[1] A widely used instance is *rectified flow* (RF) with $\alpha_t = t$ and $\beta_t = 1 - t$ (Liu et al., 2023).

**Training objective.** FM is trained with the *conditional flow matching* loss (Lipman et al., 2023)

$$\mathcal{L}_{\mathrm{CFM}}(\theta) = \mathbb{E}_{t \sim \mathcal{U}[0,1]} \mathbb{E}_{\substack{\overline{X}_0 \sim \pi_0 \\ \overline{X}_1 \sim \pi_1}} \left[ \left\| v_\theta(\overline{X}_t, t) - u_t(\overline{X}_t \mid \overline{X}_0, \overline{X}_1) \right\|_2^2 \right], \tag{3}$$

which regresses the pathwise velocity toward its conditional mean at uniformly sampled times.

**Sampling.** After training, sample $X_0 \sim \pi_0$ and evolve the learned flow by solving the ODE

$$dX_t = v_\theta(X_t, t) \, dt, \tag{4}$$

which produces a path $(X_t)_{t \in [0,1]}$ whose marginals match those of the reference path $(\overline{X}_t)_{t \in [0,1]}$ under standard existence–uniqueness conditions; in particular $X_1 \sim \pi_1$ Lipman et al. (2023). More generally, FM admits a stochastic formulation (Domingo-Enrich et al., 2025) in which the drift is augmented by a arbitrary schedule-dependent correction with diffusion coefficient $\sigma(t) \geq 0$:

$$dX_t = \underbrace{\left( v_\theta(X_t, t) + \frac{\sigma(t)^2}{2\beta_t \left( \frac{\dot{\alpha}_t}{\alpha_t} \beta_t - \dot{\beta}_t \right)} \left( v_\theta(X_t, t) - \frac{\dot{\alpha}_t}{\alpha_t} X_t \right) \right)}_{:=b(X_t, t)} dt + \sigma(t) \, dB_t, \tag{5}$$

where $(B_t)_{t \geq 0}$ is standard Brownian motion in $\mathbb{R}^d$. We will refer to $b(X_t, t)$ as the (base) *drift*. Setting $\sigma \equiv 0$ recovers the deterministic ODE.

**Connection to denoising diffusion.** Classical denoising diffusion models arise as special cases of FM when their discrete procedures are lifted to continuous time; refer to Appendix C for details.

## 3 METHODOLOGY

We formulate disentanglement as optimal control over flow-matching dynamics, derive single-pass test-time and fine-tuned controllers, and introduce a probabilistic attention loss, FOCUS.

### 3.1 STOCHASTIC OPTIMAL CONTROL

Our goal is to reduce multi-subject entanglement while remaining close to the base model. To this end, we introduce a small *control* $u : \mathbb{R}^d \times [0, 1] \to \mathbb{R}^d$ into the drift and pose generation as a quadratic, control-affine SOC problem:

$$\min_{u \in \mathcal{U}} \mathbb{E} \left[ \int_0^1 \frac{1}{2} \| u(X_t^u, t) \|_2^2 + f(X_t^u, t) dt + g(X_1^u) \right], \tag{6}$$

$$\text{s.t.} \quad dX_t^u = (b(X_t^u, t) + \sigma(t) u(X_t^u, t)) \, dt + \sigma(t) dB_t, \qquad X_0^u \sim \pi_0, \tag{7}$$

where $X_t^u$ is the latent state, $b$ is the base FM drift, $\sigma(t) \geq 0$ is a scalar diffusion schedule, and $(B_t)_{t \in [0,1]}$ is Brownian motion. The running cost $f : \mathbb{R}^d \times [0, 1] \to \mathbb{R}$ will measure subject entanglement (e.g. $f \equiv$ FOCUS), and we set the terminal cost $g \equiv 0$ in all derivations and experiments.

For control-affine dynamics with $\ell(x, u, t) = \frac{1}{2} \| u \|_2^2 + f(x, t)$, the Hamiltonian of the SOC is

$$\mathcal{H}(x, u, a, t) = \frac{1}{2} \| u \|_2^2 + f(x, t) + a^\top \left( b(x, t) + \sigma(t) u \right), \tag{8}$$

---

[1]Over-dot denotes the time derivative, i.e., $\dot{x}_t = \frac{d}{dt} x_t$.

where $a(t) \in \mathbb{R}^d$ is the co-state (adjoint). Since $\mathcal{H}$ is strictly convex in $u$, the first-order optimality condition yields

$$u_t^\star = -\sigma(t)a(t), \tag{9}$$

with adjoint dynamics

$$\frac{d}{dt}a(t) = -\left[\nabla_X b(X_t^u, t)^\top a(t) + \nabla_X f(X_t^u, t)\right], \qquad a(1) = \nabla_X g(X_1^u). \tag{10}$$

### 3.2 ON-THE-FLY DISENTANGLEMENT (TEST-TIME CONTROL)

At inference, we solve Equation (6) *per trajectory* with frozen model parameters. The idea is to compute $u_t^\star$ on-the-fly and steer the sampling process at each timestep $t$. Directly computing $u_t^\star$ requires the adjoint $a(t)$ in Equation (9), which is defined along the *controlled* path via Equation (10). This is impractical because $a(t)$ depends on the terminal condition $a(1) = \nabla_X g(X_1^u)$, which depends on the endpoint $X_1^u$, which in turn depends on the future segment $(X_\tau^u)_{\tau \in [t,1]}$; coupling a backward solve to the forward pass at every step. To obtain a *single-pass* controller, we approximate $a(t)$ locally at the current state. Concretely, we linearize Equation (10) around $X_t^u$, freeze $\nabla_X b \approx 0$, and treat the future state as locally constant:

$$a(t) \approx \int_t^1 \nabla_X f(X_t^u, \tau) d\tau \approx (1-t)\nabla_X f(X_t^u, t), \tag{11}$$

where the last step uses a left-Riemann approximation. Substituting into Equation (9) yields the instantaneous control

$$u_t^\star \approx -\sigma(t)(1-t)\nabla_X f(X_t^u, t). \tag{12}$$

The approximation $\nabla_X b \approx 0$ is common in online control settings (Havens et al., 2025).

**Velocity reparameterization (SDE).** Let $v_{\text{base}}$ denote the base FM velocity. We adopt the *memoryless* diffusion schedule, which makes the stochastic interpolant endpoints independent ($X_0 \perp X_1$) and yields a simple drift–velocity identity:

$$\sigma_{\text{mem}}(t) = \sqrt{2\,\beta_t\left(\frac{\dot{\alpha}_t}{\alpha_t}\beta_t - \dot{\beta}_t\right)}. \tag{13}$$

Under this choice, the drift-velocity relation from Equation (5) simplifies to $b(X_t, t) = 2v_\theta(X_t, t) - \frac{\dot{\alpha}_t}{\alpha_t}X_t$. Adding $+\sigma_{\text{mem}}(t)u_t$ to the drift shifts the velocity by $+\frac{1}{2}\sigma_{\text{mem}}(t)u_t$. Therefore the controlled velocity is

$$v_t^\star = v_{\text{base}}(X_t, t) + \frac{\sigma_{\text{mem}}(t)}{2}u_t^\star \approx v_{\text{base}}(X_t, t) - \frac{\sigma_{\text{mem}}^2(t)}{2}(1-t)\nabla_X f(X_t, t), \tag{14}$$

which can be passed to any SDE solver without modifying the integrator.[2]

**Deterministic alternative (ODE).** Many off-the-shelf T2I models are optimized for ODE sampling ($\sigma \equiv 0$). Decoupling $\sigma$ from the control gives

$$\min_u \mathbb{E}\left[\int_0^1 \frac{1}{2}\|u(X_t, t)\|_2^2 + f(X_t, t)dt\right] \tag{15}$$

$$\text{s.t.} \quad dX_t = (v_{\text{base}}(X_t, t) + u(X_t, t))\,dt, \quad X_0 \sim \pi_0. \tag{16}$$

The Hamiltonian $\mathcal{H} = \frac{1}{2}\|u\|^2 + f + a^\top(v_{\text{base}} + u)$ yields $u_t^\star = -a(t)$, and with the same local approximation:

$$v_t^\star = v_{\text{base}}(X_t, t) - a(t) \approx v_{\text{base}}(X_t, t) - (1-t)\nabla_X f(X_t, t), \tag{17}$$

---

[2]If desired, the factor $\frac{1}{2}\sigma_{\text{mem}}(t)^2$ can be absorbed into the weight of $f$, yielding a schedule-invariant update.

### 3.3 FINE-TUNING FOR DISENTANGLEMENT

Our goal is to learn a control network $u_\theta$ that remains close to the base dynamics and generalizes beyond the specific trajectories used during training.

**Adjoint Matching (AM).** Directly solving Equation (10) during training is prohibitive because the adjoint $a(t)$ depends on the controlled path $X_t^u$. Instead, we use *Adjoint Matching* (Domingo-Enrich et al., 2025), regressing $u_\theta$ to a cheaper *lean adjoint* $\tilde{a}$ computed along frozen forward trajectories $(X_t)_{t\in[0,1]}$ while dropping $u$-dependent Jacobian terms:

$$\frac{d}{dt}\tilde{a}(t) = -\left[\nabla_X b(X_t,t)^\top \tilde{a}(t) + \nabla_X f(X_t,t)\right], \qquad \tilde{a}(1) = \nabla_X g(X_1). \tag{18}$$

**Memoryless training.** To ensure that the learned control generalizes beyond the specific trajectories used in training, we follow Domingo-Enrich et al. (2025) and train under a *memoryless* generative process where $X_0 \perp X_1$, i.e., $p(X_0, X_1) = p(X_0)p(X_1)$. For linear (Gaussian) FM paths with scheduler $(\alpha_t, \beta_t)_{t\in[0,1]}$, the diffusion coefficient $\sigma_{\text{mem}}$ from Equation (13) achieves this independence and makes the regression target *trajectory-stationary*.

**Training objective.** Each iteration proceeds as follows: (i) sample forward trajectories $(X_t)_{t\in[0,1]}$ under $\sigma_{\text{mem}}$ with the current model frozen via Equation (5); (ii) integrate the lean adjoint $(\tilde{a}(t))_{t\in[0,1]}$ backward with Equation (18); (iii) regress the control toward the stationary target $-\sigma_{\text{mem}}(t)\tilde{a}(t)$ by minimizing

$$\mathcal{L}_{\text{AM}}(\theta) := \frac{1}{2}\int_0^1 \|u_\theta(X_t,t) + \sigma_{\text{mem}}\tilde{a}(t)\|^2 dt. \tag{19}$$

The memoryless schedule is only required during *fine-tuning*. At inference $\sigma(t)$ can be set to zero, allowing to use faster off-the-shelf ODE samplers.

### 3.4 MEASURING MULTI-SUBJECT ENTANGLEMENT

At each sampling step, T2I backbones compute *cross-attention* from image-space queries to text tokens. Empirically, these token-wise cross-attention maps correlate with the eventual spatial placement of the corresponding entities (Hertz et al., 2023; Chefer et al., 2023). This enables us to diagnose and mitigate subject entanglement *during* generation by measuring spatial interactions among *subject-specific* attention maps, rather than relying solely on post-hoc image encoders (see Figure 2).

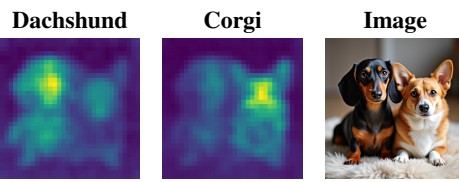

**Dachshund**     **Corgi**     **Image**

*"A dachshund and a corgi sitting together on a cozy rug"*

Figure 2: Extracted cross-attention maps for both subjects in FLUX.1 [dev].

Most prior work that optimizes multi-subject behavior via cross-attention treats these maps as generic similarity scores (e.g., maximizing cosine similarity (Meral et al., 2024) or activation differences (Chefer et al., 2023)). However, cross-attention arises from a softmax: each map is a *probability distribution* over spatial locations. Ignoring this structure discards a principled probabilistic footing and can induce artifacts such as over-concentration. We instead treat attention maps as distributions and optimize them accordingly.

**FOCUS.** Let $d$ denote the number of spatial locations and let $\Delta^{d-1}$ be the probability simplex. For a finite set $P = \{vp_1, \ldots, \boldsymbol{p}_n\} \subset \Delta^{d-1}$ of distributions, define the Jensen–Shannon divergence

$$D_{\text{JS}}(P) = \frac{1}{n}\sum_{i=1}^n D_{\text{KL}}(\boldsymbol{p}_i\|\boldsymbol{m}), \qquad \boldsymbol{m} = \frac{1}{n}\sum_{j=1}^n \boldsymbol{p}_j,$$

with $D_{\text{KL}}(\boldsymbol{p}\|\boldsymbol{q}) = \sum_{i=1}^d p_i \log \frac{p_i}{q_i}$ being the Kullback-Leibler divergence. Since $D_{\text{JS}}(P) \in [0, \log n]$, we normalize by dividing with $\log n$ to obtain $\widehat{D}_{\text{JS}}(P) \in [0, 1]$, which makes scores comparable across different set sizes; see Theorem B.1 for a proof of this upper bound.

We introduce FOCUS (Flow Optimal Control for Unentangled Subjects) to encourage, for each subject, *unimodal, spatially localized, and nonoverlapping* attention. Let $S$ be the set of subjects in

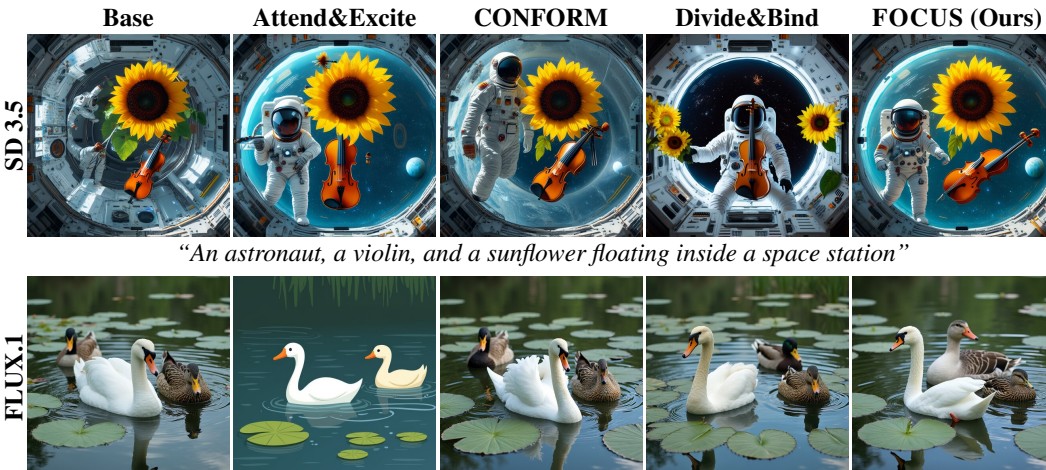

| Base | Attend&Excite | CONFORM | Divide&Bind | FOCUS (Ours) |

*"An astronaut, a violin, and a sunflower floating inside a space station"*

*"A swan, a goose, and a duck drifting past lily pads"*

Figure 3: Qualitative results with test-time control on Stable Diffusion 3.5 and FLUX.1. Each heuristic is shown at its optimal $\lambda$. Additional examples appear in Figures 9 and 10 of the Appendix.

the prompt. For each subject $s \in S$, collect its attention maps at the current step into $P_s \subset \Delta^{d-1}$ (e.g., across layers or heads), and define the subject mean $m_s = \frac{1}{|P_s|} \sum_{p \in P_s} p$. Let $M = \{m_s \mid s \in S\}$ be the set of subject means. Our FOCUS loss combines *within-subject agreement* and *between-subject separation*:

$$\text{FOCUS}(S) = \frac{1}{2}\left(\frac{1}{|S|}\sum_{s \in S}\widehat{D}_{\text{JS}}(P_s)\right) + \frac{1}{2}\left(1 - \widehat{D}_{\text{JS}}(M)\right) \tag{20}$$

The first term penalizes dispersion within each subject's maps (encouraging consistent binding, and for multi-encoder models such as SD 3.5, agreement across encoders). The second term rewards separation among subjects by maximizing divergence between their mean attention distributions. By construction, focus $\in [0, 1]$: 0 indicates perfect disentanglement (low intra-subject dispersion and maximal inter-subject separation), while larger values indicate greater entanglement.

## 4 RELATED WORK

We review approaches to *multi-subject* T2I generation. We first cover *training-free* attention-space interventions that operate at inference time. We then discuss methods that enforce *regional/layout* constraints or combine multiple diffusion paths. Finally, we survey *training-time* objectives that strengthen subject–attribute binding.

**Attention-space interventions (training-free).** A large body of work steers pre-trained generators at inference by manipulating cross-attention. At each sampling step, the model produces attention weights from spatial queries to text tokens; selecting the column for a token and normalizing over space yields a token-conditioned spatial map. Methods then *assess* entanglement (e.g., by measuring overlap across subjects) and *modify* attention or latents to promote coverage and separation.

*Attend&Excite* amplifies token activations to enforce entity coverage and reduce neglect or leakage (Chefer et al., 2023). *Divide&Bind* adds an inference-time objective that separately enforces subject coverage and attribute binding, optimizing latents during sampling (Li et al., 2023b). *Structured Diffusion Guidance* injects linguistic structure (e.g., dependency trees) to guide attention manipulation for multi-object composition (Feng et al., 2023). *Prompt-to-Prompt* locks cross-attention correspondences to preserve word–subject alignments across edits, often used to maintain multi-subject layouts (Hertz et al., 2023). *CONFORM* formulates a contrastive, InfoNCE-style objective that separate different subjects while pulling subject–attribute pairs together (Meral et al., 2024).

While effective in specific setups, these methods are heuristic and lack a unifying optimization principle; moreover, many were developed for Stable Diffusion 1.x backbones, limiting portability to modern flow-matching models. In contrast, our method derives a controller from a single SOC

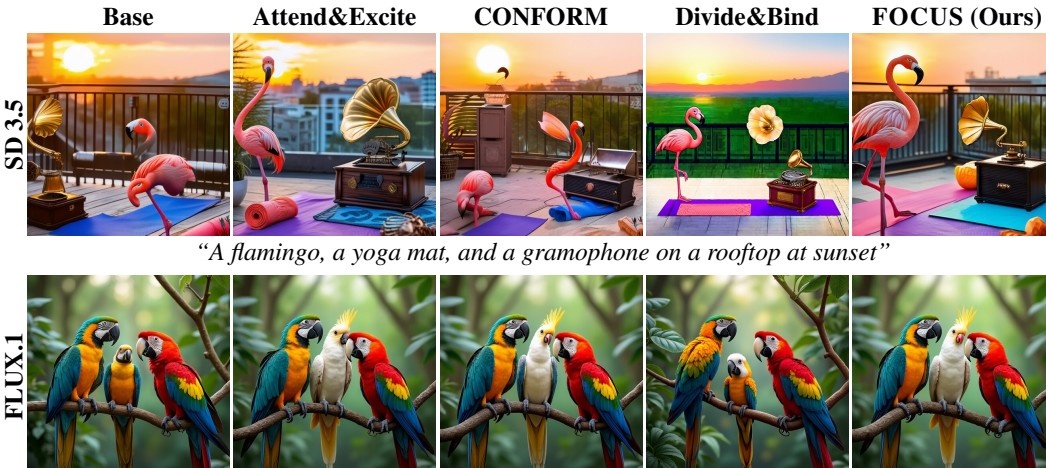

| Base | Attend&Excite | CONFORM | Divide&Bind | FOCUS (Ours) |
|---|---|---|---|---|

*"A flamingo, a yoga mat, and a gramophone on a rooftop at sunset"*

*"A macaw, a cockatoo, and an Amazon parrot perched on a jungle vine"*

Figure 4: Qualitative results after fine-tuning Stable Diffusion 3.5 and FLUX.1. Each heuristic uses its optimal hyperparameters. Additional examples appear in Figures 11 and 12 of the Appendix.

objective at the FM level, yielding architecture-agnostic updates. We also instantiate our controller with costs derived from several of the above heuristics to demonstrate principled portability.

**Regional/layout composition and multi-path fusion.** A complementary direction constrains *where* subjects appear. *MultiDiffusion* fuses multiple diffusion trajectories under shared spatial constraints (e.g., boxes or masks), enabling faithful multi-subject placement without retraining (Bar-Tal et al., 2023). Related systems extend this idea to interactive, region-based workflows. *GLIGEN* augments a frozen backbone with grounding layers and conditions on bounding boxes or phrases to place multiple objects precisely (Li et al., 2023a). More recently, *Be Decisive* leverages the layout implicitly encoded in the initial noise and refines it during denoising, avoiding conflicts with externally imposed layouts and improving prompt alignment while preserving model priors (Dahary et al., 2025). These approaches disentangle primarily via spatial decoupling but often require user-specified or learned layouts, which increases user effort and restricts spontaneous subject interaction. Our method reduces entanglement without explicit spatial annotations, requiring only the text prompt (and its subjects).

**Training-time objectives for multi-subject fidelity.** Some works alter training signals to strengthen subject–attribute binding. *TokenCompose* introduces token-level supervision to improve consistency for prompts with multiple categories and attributes (Wang et al., 2024b). Region-aware objectives decompose complex prompts into per-region descriptions and enforce alignment, reducing cross-entity leakage. Such methods typically assume curated supervision and substantial retraining. In contrast, our fine-tuning objective is lightweight: it adapts pre-trained models via Adjoint Matching and requires only text prompts, while our test-time controller operates with zero parameter updates.

## 5 EXPERIMENTS

We evaluate our approach in three stages. We first describe datasets, metrics, models, and baselines. We then present *test-time* (on-the-fly) results, followed by *fine-tuning* results. All experiments ran on NVIDIA A100/H100 GPUs. While the test-time controller runs on commodity GPUs with as little as 12 GB VRAM, fine-tuning experiments fit within the VRAM of H100 GPUs.

**Base Models.** We report results on two open-source flow-matching models: *Stable Diffusion 3.5* (SD 3.5) (Esser et al., 2024) and *FLUX.1 [dev]* (FLUX.1) (Labs et al., 2025).

**Dataset.** We create a 150-prompt corpus with 2–4 subjects per prompt using GPT-5. Half the prompts contain *similar* subjects (e.g., "a black bear and a brown bear[...]"); the rest contain *dissimilar* subjects (e.g., "a snowboard, a telescope, and a husky[...]"). For each prompt, we annotate subject token indices for both CLIP and T5 encoders to extract cross-attention maps for the heuristics. Such per-subject annotations are typically absent from existing corpora.

Table 1: Test-time control results at the optimal $\lambda$ for each heuristic. We report mean$\pm$ std over all prompts and seeds; the top three per metric are highlighted (gold/silver/bronze).

| | Heuristic | CLIP I-T↑ | SigLIP-2 I-T↑ | BLIP T-T↑ | Qwen2 T-T↑ | PickScore I-T↑ | ImgRew I-T↑ | Composite↑ |
|---|---|---|---|---|---|---|---|---|
| **SD 3.5** | Base | $0.3474_{\pm0.03}$ | $0.2309_{\pm0.05}$ | $0.5731_{\pm0.15}$ | $0.6402_{\pm0.08}$ | $22.6940_{\pm0.99}$ | $1.3175_{\pm0.68}$ | $0.0000_{\pm0.00}$ |
| | Attend&Excite | $0.3484_{\pm0.03}$ | $0.2326_{\pm0.05}$ | $0.5752_{\pm0.15}$ | $0.6404_{\pm0.08}$ | $22.6950_{\pm1.01}$ | $1.3545_{\pm0.66}$ | $3.1714_{\pm0.53}$ |
| | CONFORM | $0.3481_{\pm0.03}$ | $0.2323_{\pm0.05}$ | $\mathbf{0.5773}_{\pm0.15}$ | $\mathbf{0.6421}_{\pm0.08}$ | $22.7188_{\pm0.99}$ | $1.3684_{\pm0.64}$ | $3.4336_{\pm0.45}$ |
| | Divide&Bind | $\mathbf{0.3489}_{\pm0.03}$ | $0.2316_{\pm0.05}$ | $0.5742_{\pm0.14}$ | $0.6399_{\pm0.08}$ | $22.6779_{\pm1.03}$ | $1.3493_{\pm0.67}$ | $3.9373_{\pm0.78}$ |
| | FOCUS (Ours) | $0.3483_{\pm0.03}$ | $\mathbf{0.2344}_{\pm0.04}$ | $0.5751_{\pm0.15}$ | $0.6385_{\pm0.08}$ | $\mathbf{22.7499}_{\pm1.02}$ | $\mathbf{1.4003}_{\pm0.62}$ | $\mathbf{4.2865}_{\pm0.86}$ |
| **FLUX.1** | Base | $0.3449_{\pm0.03}$ | $0.2271_{\pm0.05}$ | $0.5739_{\pm0.15}$ | $0.6300_{\pm0.09}$ | $23.4234_{\pm1.03}$ | $\mathbf{1.2970}_{\pm0.66}$ | $0.0000_{\pm0.00}$ |
| | Attend&Excite | $0.3430_{\pm0.03}$ | $0.2242_{\pm0.05}$ | $0.5716_{\pm0.14}$ | $0.6304_{\pm0.09}$ | $23.2549_{\pm1.11}$ | $1.2494_{\pm0.70}$ | $1.7595_{\pm0.67}$ |
| | CONFORM | $0.3436_{\pm0.03}$ | $0.2252_{\pm0.05}$ | $0.5726_{\pm0.15}$ | $0.6321_{\pm0.09}$ | $23.3574_{\pm1.03}$ | $1.2461_{\pm0.70}$ | $1.5114_{\pm0.26}$ |
| | Divide&Bind | $\mathbf{0.3453}_{\pm0.03}$ | $\mathbf{0.2272}_{\pm0.05}$ | $0.5722_{\pm0.15}$ | $\mathbf{0.6330}_{\pm0.08}$ | $\mathbf{23.4395}_{\pm1.02}$ | $1.2939_{\pm0.67}$ | $1.6352_{\pm0.44}$ |
| | FOCUS (Ours) | $0.3446_{\pm0.03}$ | $0.2268_{\pm0.05}$ | $\mathbf{0.5741}_{\pm0.14}$ | $0.6326_{\pm0.08}$ | $23.4274_{\pm1.02}$ | $1.2913_{\pm0.67}$ | $\mathbf{1.9712}_{\pm0.31}$ |

**Metrics.** To quantify multi-subject fidelity, we follow Yu & Chien (2025) and report two alignment groups. For image–text (I–T) alignment we compute CLIP (Radford et al., 2018) and SigLIP-2 (Tschannen et al., 2025) cosine similarities between image and prompt embeddings. For caption-based text–text (T–T) faithfulness, we caption each image with BLIP (Li et al., 2022) and Qwen2-VL (Wang et al., 2024a) and measure semantic similarity to the prompt. We additionally report preference-trained scores, PickScore (Kirstain et al., 2023) and ImageReward (Xu et al., 2023), as proxies for human preference.

For model selection, we compute a composite score per hyperparameter combination by *macro-averaging* baseline-relative gains across metrics; see Appendix D.2 for the formula. Because we aim to preserve base style and subject depiction, global alignment scores may shift modestly even when multi-subject fidelity improves. Unless noted otherwise, we generate five images per prompt (distinct seeds) per hyperparameter setting, fixing sampler, steps, guidance, and resolution allowing to make direct comparisons between test-time control and fine-tuning. Full details are in Appendix D.

**Baselines and heuristics.** To demonstrate portability across FM models and legacy U-Net heuristics, we evaluate *Attend&Excite* (Chefer et al., 2023), *CONFORM* (Meral et al., 2024), *Divide&Bind* (Li et al., 2023b), and our heuristic FOCUS. Because cost magnitudes differ, we optimize a scaled running cost $\lambda \cdot f(X_t, t)$ with $\lambda > 0$. The effect of $\lambda$ at test time is shown in Figure 7.

**Human study.** Automated metrics struggle to detect attribute leakage reliably (Dahary et al., 2025), so we conducted a prompt-conditioned, pairwise preference study with 50 participants. In each trial, participants viewed two images from our evaluation suite alongside the prompt and selected the image that better matched the prompt, yielding 2,000 pairwise judgments. From these outcomes we computed Elo ratings (across-method comparability) and win rates (fraction of pairwise wins).

## 5.1 ON-THE-FLY DISENTANGLEMENT (TEST-TIME CONTROL)

We sweep ten $\lambda$ values per heuristic and select the best via the composite score defined above. Table 1 report per-heuristic, per-model results at the optimal $\lambda$, qualitative examples are shown in Figures 9 and 10, and Human Study results are summarized in Table 2.

All heuristics outperform the base sampler on SD 3.5 and FLUX.1, indicating that the SOC formulation yields a principled route to port legacy heuristics to modern FM models. Qualitatively, outputs show higher multi-subject fidelity: subjects are more often present and better separated than in the base model. Our human study shows similar trends, with higher win-rates and Elo ratings. While FOCUS is not best on every metric, it attains the highest composite score across all heuristics and achieves almost all best scores in our human study.

Table 2: Human preference study for test-time control. Report pairwise win rate and Elo rating; see appendix E for details.

| Heuristic | SD3.5 | | FLUX.1 | |
|---|---|---|---|---|
| | Win[%] | Elo↑ | Win[%] | Elo↑ |
| Base | 45% | 1517 | 46% | 1464 |
| Attend&Excite | 53% | 1500 | 49% | 1526 |
| CONFORM | 42% | 1373 | 50% | 1498 |
| Divide&Bind | 50% | 1562 | 50% | 1450 |
| FOCUS (Ours) | **58%** | 1548 | **54%** | **1562** |

## 5.2 FINE-TUNING FOR DISENTANGLEMENT

We insert LoRA layers (Hu et al., 2022) into self-attention blocks and freeze all base parameters. We use rank $r=4$ (training $< 0.1\%$ of parameters). We sweep $\lambda$ and other hyperparameters, including

Table 4: Fine-tuning results at the set of hyperparameters for each heuristic. We report mean± std over all prompts and seeds; the top three per metric are highlighted (gold/silver/bronze).

| | Heuristic | CLIP I-T↑ | SigLIP-2 I-T↑ | BLIP T-T↑ | Qwen2 T-T↑ | PickScore I-T↑ | ImgRew I-T↑ | Composite↑ |
|---|---|---|---|---|---|---|---|---|
| **SD 3.5** | Base | $0.3474_{\pm0.03}$ | $0.2309_{\pm0.05}$ | $0.5731_{\pm0.15}$ | $0.6402_{\pm0.08}$ | $22.6940_{\pm0.99}$ | $1.3175_{\pm0.68}$ | $0.0000_{\pm0.00}$ |
| | Attend&Excite | $0.3469_{\pm0.03}$ | $0.2281_{\pm0.04}$ | $0.5747_{\pm0.15}$ | $\mathbf{0.6425}_{\pm0.08}$ | $\mathbf{22.8429}_{\pm1.01}$ | $1.4460_{\pm0.60}$ | $5.7181_{\pm1.21}$ |
| | CONFORM | $0.3478_{\pm0.03}$ | $0.2294_{\pm0.05}$ | $0.5646_{\pm0.15}$ | $0.6393_{\pm0.09}$ | $22.5962_{\pm0.99}$ | $1.3782_{\pm0.63}$ | $3.4583_{\pm1.05}$ |
| | Divide&Bind | $0.3486_{\pm0.03}$ | $0.2266_{\pm0.05}$ | $\mathbf{0.5870}_{\pm0.14}$ | $0.6358_{\pm0.08}$ | $22.3401_{\pm0.99}$ | $1.3524_{\pm0.68}$ | $0.8006_{\pm0.69}$ |
| | FOCUS (Ours) | $\mathbf{0.3495}_{\pm0.03}$ | $\mathbf{0.2331}_{\pm0.04}$ | $0.5744_{\pm0.15}$ | $0.6383_{\pm0.08}$ | $22.6445_{\pm0.97}$ | $\mathbf{1.4495}_{\pm0.58}$ | $5.9174_{\pm1.19}$ |
| **FLUX.1** | Base | $0.3449_{\pm0.03}$ | $0.2271_{\pm0.05}$ | $0.5739_{\pm0.15}$ | $0.6300_{\pm0.09}$ | $\mathbf{23.4234}_{\pm1.03}$ | $1.2970_{\pm0.66}$ | $0.0000_{\pm0.00}$ |
| | Attend&Excite | $\mathbf{0.3468}_{\pm0.03}$ | $0.2320_{\pm0.05}$ | $\mathbf{0.5876}_{\pm0.15}$ | $0.6382_{\pm0.08}$ | $23.3333_{\pm1.01}$ | $1.3806_{\pm0.62}$ | $2.3477_{\pm0.79}$ |
| | CONFORM | $0.3458_{\pm0.03}$ | $0.2305_{\pm0.04}$ | $0.5800_{\pm0.15}$ | $0.6369_{\pm0.08}$ | $23.3724_{\pm1.00}$ | $1.3631_{\pm0.63}$ | $1.9591_{\pm0.83}$ |
| | Divide&Bind | $0.3445_{\pm0.03}$ | $0.2296_{\pm0.05}$ | $0.5705_{\pm0.15}$ | $0.6246_{\pm0.09}$ | $23.1909_{\pm1.06}$ | $1.2269_{\pm0.70}$ | $0.2002_{\pm0.47}$ |
| | FOCUS (Ours) | $0.3468_{\pm0.03}$ | $\mathbf{0.2328}_{\pm0.05}$ | $0.5780_{\pm0.15}$ | $\mathbf{0.6386}_{\pm0.08}$ | $23.3278_{\pm1.01}$ | $\mathbf{1.3899}_{\pm0.61}$ | $\mathbf{2.5881}_{\pm0.79}$ |

dataset choice; see Appendix D. With the best settings, fine-tuning takes 17 min on SD 3.5 and 79 min on FLUX.1. Table 4 reports metrics across all heuristics and models; qualitative results appear in Figures 4, 11 and 12; human preferences are summarized in Table 3.

Training data comprise two small subsets of our evaluation dataset. The first is a single prompt, "A horse and a bear in a forest," where SD 3.5 fails reliably (HORSE&BEAR). The second contains 15 prompts with two semantically similar subjects (TWOOBJECTS). Despite their size, both subsets yield gains on diverse *unseen* prompts, including different subject categories, prompts with more than two subjects, and different subject token positions, suggesting that our method targets a core failure mode in multi-subject composition.

Table 3: Human preference study for fine-tuned models. Report pairwise win rate and Elo rating; see appendix E for details.

| Heuristic | SD3.5 | | FLUX.1 | |
|---|---|---|---|---|
| | Win[%] | Elo↑ | Win[%] | Elo↑ |
| Base | 39% | 1355 | 51% | 1462 |
| Attend&Excite | 56% | 1584 | 50% | 1476 |
| CONFORM | 49% | 1520 | 50% | **1620** |
| Divide&Bind | 48% | 1436 | 43% | 1442 |
| FOCUS (Ours) | **57%** | **1605** | **54%** | 1500 |

Across the board, the fine-tuned models outperform their test-time controlled counterparts. This matches our theory, since during fine-tuning the adjoint signal is computed explicitly over the full trajectory, whereas test-time control relies on a single-pass approximation. Between the two training sets, HORSE&BEAR yields the strongest gains with an 85% relative improvement in the composite scores in contrast to TWOOBJECTS for SD 3.5 and about 5% for FLUX.1. Across heuristics, FO-CUS attains the highest composite score, indicating the largest average improvement across metrics. Consistently, in the human study FOCUS achieves the highest win rates against competing heuristics and among the highest Elo ratings for both models.

### 5.3 CLASSICAL DENOISING DIFFUSION

Although our algorithms are derived for flow matching, Appendix C establishes a correspondence denoising diffusion. To test the theory, we apply the *test-time controller* to *Stable Diffusion XL*, a U-Net based denoising diffusion model. As shown in Figure 5, FOCUS improves on the prompt 'A lion and a tiger resting side by side[...]' by reducing attribute leakage.

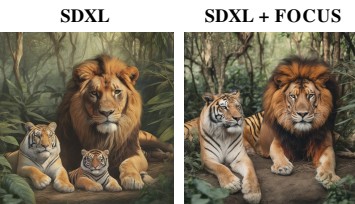

**SDXL**    **SDXL + FOCUS**

Figure 5: Transfer to SDXL.

### 6 DISCUSSION AND FUTURE WORK

We propose a control-theoretic framework for multi-subject fidelity, instantiated either as a single-pass test-time controller or as a lightweight fine-tuned controller. The formulation accommodates existing attention-based heuristics, and our FOCUS yields the most consistent gains across settings. The two realizations offer complementary trade-offs: test-time control applies directly to a frozen model given subject tokens, at the cost of roughly $2\times$ longer inference, whereas fine-tuning requires subject tokens only during training and matches the base model's inference speed during inference. Finally, the strong generalization of fine-tuning—even from a single prompt—suggests an underlying attention-level failure mode in current T2I models; future work should probe this mechanism and develop annotation-free proxies and automated subject tokenization.

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

APPENDIX CONTENTS

## A  USE OF LARGE LANGUAGE MODELS

We used a large language model (OpenAI GPT-5 via ChatGPT) for two purposes: (i) expanding a small, human-written set of text prompts to create additional prompts for our synthetic dataset, and (ii) polishing writing (grammar, clarity, and tone). For dataset construction, the model generated semantically similar prompt variants; all outputs were screened and curated by the authors. For writing, the authors drafted all sections and used the model only for copy-editing, not for introducing technical content.

## B  FOCUS

This appendix details FOCUS, our probabilistic attention heuristic used as a running cost for disentanglement. We emphasize three practical design choices: (i) encoding spatial proximity before measuring divergence, (ii) aggregating attention maps prior to scoring, and (iii) omitting an explicit collapse regularizer.

**Spatially aware divergence.** We promote separation of subjects by maximizing a Jensen–Shannon divergence (JSD) defined over attention distributions. A naïve computation on flattened maps discards locality, allowing distant activations to interact as if adjacent. To preserve spatial structure, we (i) reshape token-embedding maps to the target aspect ratio, (ii) apply a light 2D Gaussian smoothing, and only then (iii) flatten for scoring. This encodes proximity and mitigates grid-like artifacts during optimization.

**Block selection and aggregation.** Modern T2I backbones follow Diffusion Transformer designs (Peebles & Xie, 2023). Rather than computing scores *per block* and averaging their scores which can result in conflicting update directions, we first aggregate attention and then score. Concretely, we average cross-attention maps over all blocks that process text and image tokens *separately*, producing a single map per token and a consistent optimization direction. Blocks that jointly process text and image tokens are excluded from this aggregation for compatibility.

**No explicit collapse regularizer.** We experimented with an entropy-based regularizer aimed at discouraging overly concentrated (collapsed) attention. Let $H(\boldsymbol{p}) = -\sum_i \boldsymbol{p}_i \log \boldsymbol{p}_i$ denote the Shannon Entropy and $\widehat{H}(\boldsymbol{p}) = H(\boldsymbol{p})/\log d \in [0,1]$ its normalized version, where $d$ is the number of spatial locations. For each subject we form its mixture distribution $\boldsymbol{m_s}$ and added

$$\gamma_{\text{reg}} \cdot \frac{1}{|S|} \sum_{\boldsymbol{s} in S} \left(1 - \widehat{H}(\boldsymbol{m})\right), \tag{21}$$

scaling by $\gamma_{\text{reg}} > 0$ to control its effect. In our experiments, small $\gamma_{\text{reg}}$ made the term largely inactive, while larger $\gamma_{\text{reg}}$ pushed mass away from the subject rather than stabilizing it, see Figure 6 for an example. We therefore omit this term in FOCUS and rely on the probabilistic objective described above.

| Base | $\gamma_{\text{reg}} = 0$ | $\gamma_{\text{reg}} = 0.01$ | $\gamma_{\text{reg}} = 1$ | $\gamma_{\text{reg}} = 10$ |
|---|---|---|---|---|

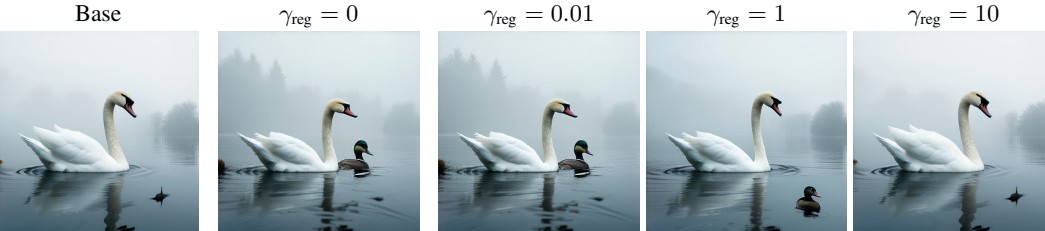

Figure 6: Ablation of regularizer strength $\gamma_{\text{reg}}$ for the test-time controller on Stable Diffusion 3.5.

**Lemma B.1** (Upper Bound of Jensen–Shannon Divergence). *Let* $P = \{\boldsymbol{p}^{(1)}, \ldots, \boldsymbol{p}^{(n)}\} \subset \Delta^{d-1}$ *be a set of probability distributions. Then,* $D_{\text{JS}}(P)$ *is upper bounded by* $\log n$.

*Proof.* Define $P$ as in theorem B.1, then the JSD is defined as follows:

$$D_{\mathrm{JS}}(P) = \frac{1}{n} \sum_{k=1}^{n} D_{\mathrm{KL}} \left( \boldsymbol{p}^{(k)} \parallel \boldsymbol{m} \right), \quad \boldsymbol{m} = \frac{1}{n} \sum_{k=1}^{n} \boldsymbol{p}^{(k)}.$$

We can upper bound each $D_{\mathrm{KL}}$-term as follows:

$$
\begin{aligned}
D_{\mathrm{KL}}(\boldsymbol{p}^{(k)} \parallel \boldsymbol{m}) &= \sum_{i=1}^{d} p_i^{(k)} \log \frac{p_i^{(k)}}{m_i} \\
&= \sum_{i=1}^{d} p_i^{(k)} \log \frac{p_i^{(k)}}{\frac{1}{n} \sum_{\ell=1}^{n} p_i^{(\ell)}} \\
&= \sum_{i=1}^{d} p_i^{(k)} \log \left( n \cdot \frac{p_i^{(k)}}{\sum_{\ell=1}^{n} p_i^{(\ell)}} \right) \\
&\leq \sum_{i=1}^{d} p_i^{(k)} \log n \\
&= \log n.
\end{aligned}
$$

Plugging this bound back into the definition of the JSD, yields the desired results:

$$\frac{1}{n} \sum_{k=1}^{n} D_{\mathrm{KL}} \left( \boldsymbol{p}^{(k)} \parallel \boldsymbol{m} \right) \leq \frac{1}{n} \sum_{k=1}^{n} \log n = \log n$$

$$\square$$

**Normalization.** Because $D_{\mathrm{JS}}(P) \in [0, \log n]$, we use the normalized score $\widehat{D}_{\mathrm{JS}}(P) = D_{\mathrm{JS}}(P)/\log n \in [0, 1]$, which makes values comparable across different set sizes.

## C  DENOISING DIFFUSION AS FLOW MATCHING

This section makes precise how classical denoising diffusion (score-based) models arise as a special case of the flow-matching (FM) framework. We first derive the continuous-time SDE limit of the variance-preserving (VP) family (Sohl-Dickstein et al., 2015; Ho et al., 2020; Song et al., 2021); after we express reverse-time generation; and finally show an explicit parameterization that uses a diffusion model as an FM velocity field. Analogous statements hold for VE and EDM variants (Nichol & Dhariwal, 2021; Karras et al., 2022; 2024).

### C.1  VP CHAIN TO SDE

Let $X_0 \sim p_{\mathrm{data}}$. The standard $K$-step VP forward noising chain is

$$X_k = \sqrt{\alpha_k} X_{k-1} + \sqrt{1 - \alpha_k} \epsilon_k, \quad \epsilon_k \sim \mathcal{N}(0, \boldsymbol{I}), \quad k = 1, \dots, K, \tag{22}$$

where $\alpha_k := 1 - \beta_k \in (0, 1)$ with $\beta_k \in (0, 1)$ typically increasing over $k$ (Ho et al., 2020). This yields the closed-form marginal

$$X_k \mid X_0 \sim \mathcal{N}\left( \sqrt{\bar{\alpha}_k}\, X_0, \, (1 - \bar{\alpha}_k)\, \boldsymbol{I} \right), \qquad \bar{\alpha}_k := \prod_{i=1}^{k} \alpha_i. \tag{23}$$

For sufficiently large $K$, $X_K$ is approximately standard normal (Ho et al., 2020).

We lift this formulation to continuous time by defining a uniform grid $\tau_k := k/K$ on $[0, 1]$, so every increment is $\Delta\tau = 1/K$. Define a piecewise-constant rate $\beta(\tau)$ via $\beta(\tau) := \beta_k/\Delta\tau$ for $\tau \in [\tau_{k-1}, \tau_k)$. Then by using the first-order Taylor approximation of $\sqrt{1+x}$, we can rewrite

$\sqrt{\alpha_k} = \sqrt{1 - \beta_k} \approx 1 - \frac{1}{2}\beta_k + \mathcal{O}(\beta_k^2)$, and obtain

$$\Delta X_k := X_k - X_{k-1} \tag{24}$$

$$= -\frac{1}{2}\beta_k X_{k-1} + \sqrt{\beta_k}\epsilon_k + \mathcal{O}(\beta_k^2) \tag{25}$$

$$= \left(-\frac{1}{2}\beta(\tau_{k-1})X_{k-1}\right)\Delta\tau + \sqrt{\beta(\tau_{k-1})}\sqrt{\Delta\tau}\epsilon_k + \mathcal{O}\left((\Delta\tau)^{\frac{3}{2}}\right). \tag{26}$$

This is the Euler–Maruyama discretizations of the forward/diffusion VP-SDE:

$$dX_\tau = -\frac{1}{2}\beta(\tau)X_\tau d\tau + \sqrt{\beta(\tau)}dB_\tau, \quad \tau \in [0, 1], \tag{27}$$

and the discrete chain converges to this SDE as $K \to \infty$. Moreover, the SDE has Gaussian marginals

$$X_\tau \mid X_0 \sim \mathcal{N}\left(\sqrt{\bar{\alpha}}(\tau)X_0, (1 - \bar{\alpha}(\tau))\,\boldsymbol{I}\right), \quad \text{with} \quad \bar{\alpha}(\tau) := \exp\left(-\int_0^\tau \beta(u)du\right), \tag{28}$$

which matches Equation (23) at the grid points if we choose $\bar{\alpha}(\tau_k) = \bar{\alpha}_k$ (Song et al., 2021).

## C.2 REVERSE-TIME DYNAMICS

We now reverse time to generate from noise to data. Let $\bar{\tau} = 1 - \tau$ denote the *generative time*. By classical time reversal diffusion (Anderson, 1982) the reverse-time process satisfies

$$dX_\tau = \left(-\frac{1}{2}\beta\tau X_\tau - \beta(\tau)\nabla_X \log p_\tau(X_\tau)\right)d\bar{\tau} + \sqrt{\beta(\tau)}d\bar{B}_\tau, \quad \text{with} \quad d\bar{\tau} = -d\tau, \tag{29}$$

where $p_\tau$ are the forward-time marginals and $\nabla_X \log p_\tau$ is the score (Song et al., 2021).

In practice, most diffusion architectures parameterize the model via *noise prediction* $\epsilon_\theta$ (Ho et al., 2020; Karras et al., 2022; 2024), which is related to the score by:

$$\nabla_X \log p_\tau(x) = -\frac{\epsilon_\theta(x, \tau)}{\sqrt{1 - \bar{\alpha}(\tau)}}. \tag{30}$$

## C.3 TIME CHANGE TO FM

To embed VP diffusion into FM, we reparameterize time so that FM runs from noise to data, setting $t := 1 - \tau$, which yields the following FM schedules:

$$\alpha_t^{\text{FM}} := \sqrt{\bar{\alpha}(1 - t)}, \quad \text{and} \quad \beta_t^{\text{FM}} := \sqrt{1 - \bar{\alpha}(1 - t)}. \tag{31}$$

## C.4 SCORE RELATIONS

For linear Gaussian reference paths, the score $s(x, t) := \nabla_X \log p_t(x)$ and the FM vector field $v_\theta(x, t)$ are linked by a schedule-dependent affine map (Lipman et al., 2023; Albergo et al., 2023; Liu et al., 2023):

$$s(x, t) = \frac{1}{\eta_t}\left(v_\theta(x, t) - \kappa_t\,x\right), \qquad \kappa_t := \frac{\dot{\alpha}_t^{\text{FM}}}{\alpha_t^{\text{FM}}}, \quad \eta_t := \beta_t^{\text{FM}}\left(\frac{\dot{\alpha}_t^{\text{FM}}}{\alpha_t^{\text{FM}}}\beta_t^{\text{FM}} - \dot{\beta}_t^{\text{FM}}\right). \tag{32}$$

Combining the noise–score relation with the time change $\tau = 1 - t$ gives:

$$s(x, t) = \nabla_X \log p_t(x) = -\frac{\epsilon_\theta(x, 1 - t)}{\beta_t^{\text{FM}}}, \tag{33}$$

since $\beta_t^{\text{FM}} = \sqrt{1 - \bar{\alpha}(1 - t)}$. Substituting this into the score–velocity map yields the corresponding FM *velocity prediction* induced by an $\epsilon$-parameterized diffusion model:

$$v_\theta(x, t) = \kappa_t x - \eta_t \frac{\epsilon_\theta(x, 1 - t)}{\beta_t^{\text{FM}}}. \tag{34}$$

This identity lets an $\epsilon$-trained diffusion model be used directly as an FM velocity field for the VP-induced schedules above; plugging $v_\theta$ into the FM SDE recovers the reverse-time VP sampler (and setting $\sigma \equiv 0$ recovers the probability-flow/DDIM ODE) under the change of variables $t = 1 - \tau$.

# D HYPERPARAMETERS

## D.1 SAMPLING PARAMETERS

For Stable Diffusion 3.5[3] and FLUX.1 [dev][4], we follow the official sampling recommendations. Unless stated otherwise, we use the deterministic Euler scheduler with 28 inference steps for both models and generate images at $512 \times 512$ resolution. The classifier-free guidance scale is set to 4.5 for SD3.5 and 3.5 for FLUX. To ensure consistent extraction of cross-attention maps, we cap the maximum tokenized sequence length at 77 for SD3.5 and 256 for FLUX, and we verify that all prompts in our dataset fall within these limits. Models are loaded and all computations are performed in `bfloat16` to reduce memory usage.

## D.2 METRICS

To summarize each hyperparameter setting with a single scalar, we macro-average the *relative improvement* over the base model across prompts, seeds, and metrics.

Let $X_{p,s}$ be the image produced by the current setting for prompt $p \in P$ and seed $s \in S$, and let $\hat{X}_{p,s}$ be the corresponding image from the base model. Let $\mathcal{M}$ denote the set of evaluation metrics. Since in our settings all metrics are increasing, the composite score for a hyperparameter setting is the macro-average

$$\frac{1}{|S|} \sum_{s \in S} \frac{1}{|P|} \sum_{p \in P} \frac{1}{|M|} \sum_{m \in M} \frac{m(X_{p,s}) - m(\hat{X}_{p,s})}{m(\hat{X}_{p,s})}, \tag{35}$$

such that a value larger than 0 indicates an average improvement over the base model, while values smaller than 0 indicate degradation.

## D.3 TEST-TIME CONTROL

In the deterministic (ODE) variant, the single-pass update does not inherit the time–weighting $\frac{1}{2}\sigma_{\text{mem}}^2(t)$ that appears in the SDE case. Since $\sigma_{\text{mem}}(t)$ is large at early times and decays rapidly as $t \to 1$, we reintroduce this desirable early–strong / late–weak behavior in the ODE setting by reweighting the running cost:

$$f(X_t, t) = \lambda \cdot \sigma_{\text{mem}}^2(t) \cdot \text{Heuristic}(X_t), \tag{36}$$

where $\lambda > 0$ is the earlier introduced hyperparameter to account for different heuristic magnitudes. Throughout our test-time control experiments, we use this time-weighted running cost variant and sweep over $\lambda \in \{0.1, 0.5, 1, 2, 3, 4, 8, 12, 16, 32\}$. Values below 0.1 have negligible effect across heuristics, while values above 32 tend to produce artifacts (over-sharpening, texture noise) or occasional numerical instabilities (NaNs). See Figure 7 for qualitative trends.

| Base | $\lambda = 0.1$ | $\lambda = 0.5$ | $\lambda = 1$ | $\lambda = 2$ | $\lambda = 3$ | $\lambda = 4$ | $\lambda = 8$ | $\lambda = 16$ | $\lambda = 32$ |

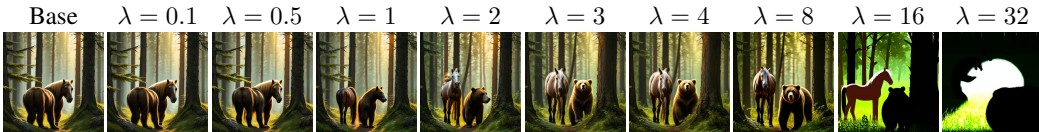

Figure 7: Effect of the control parameter $\lambda$ on test-time control with Stable Diffusion 3.5.

## D.4 FINE-TUNING

We initialize the *memoryless* schedule from each model's ODE 28-step inference schedule (same time steps), do not use classifier-free guidance, and for **FLUX.1** apply its native guidance scale (not CFG). Following Appendix D.1, we cap tokenized sequence length for cross-attention extraction to 77 (SD 3.5) and 256 (FLUX.1). Models are loaded in `bfloat16`; forward/backward passes run in

---

[3]https://huggingface.co/stabilityai/stable-diffusion-3.5-medium
[4]https://huggingface.co/black-forest-labs/FLUX.1-dev

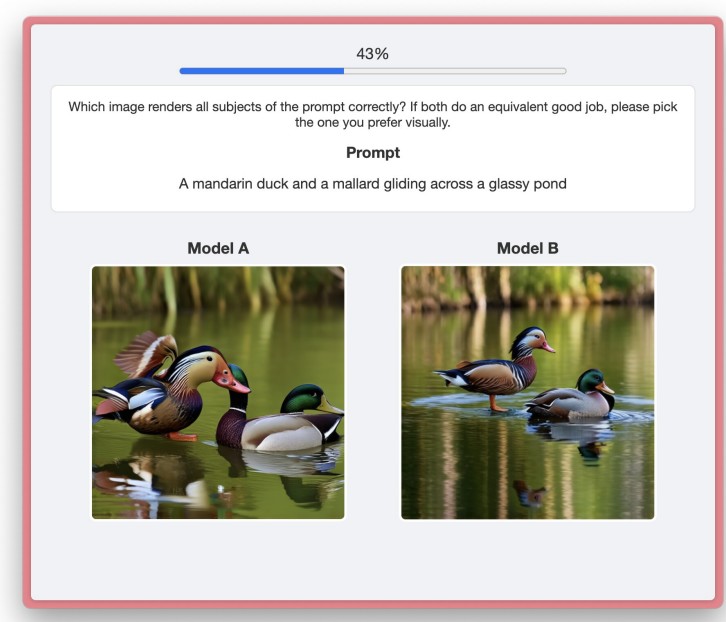

Figure 8: User interface for the prompt-conditioned, pairwise preference study.

BF16 and the final loss reduction is computed in FP32 to avoid numerical issues. To reduce memory, at each iteration we subsample 16 of the 28 steps to be used in our loss calculation. We further use a batch sizes of 5 trajectories for SD 3.5 and 2 trajectories for FLUX.1. We use two small prompt sets: HORSE&BEAR (single prompt: "A horse and a bear") and TWOOBJECTS (15 prompts, each with two semantically similar subjects). Optimization uses AdamW with a weight decay of 0.01 and $\beta_0 = 0.95$, $\beta_1 = 0.999$. In addition, we also employ `Accelerate` to lower peak memory consumption. Table 5 lists the hyperparameter grids we sweep per heuristic; best settings are **bold**.

Table 5: Hyperparameter grids for fine-tuning; best settings per row in **bold**.

| | Heuristic | Lambda $\lambda$ | Learning rate | Checkpoint | Dataset |
|---|---|---|---|---|---|
| **SD 3.5** | Attend&Excite | $\{0.1, 1, \mathbf{10}\}$ | $5e{-}5$ | $\{\mathbf{100}, 150\}$ | HORSE&BEAR |
| | CONFORM | $\{\mathbf{0.1}, 1, 10\}$ | $5e{-}5$ | $\{\mathbf{100}, 150\}$ | HORSE&BEAR |
| | Divide&Bind | $\{\mathbf{0.1}, 1, 10\}$ | $5e{-}5$ | $\{\mathbf{100}, 150\}$ | HORSE&BEAR |
| | FOCUS | $\{0.01, 0.1, \mathbf{1}, 10, 100\}$ | $\{1e{-}4, \mathbf{5e{-}5}, 1e{-}5\}$ | $\{\mathbf{100}, 150, 200\}$ | $\{\mathbf{HORSE\&BEAR}, \text{TWOOBJECT}\}$ |
| **FLUX.1** | Attend&Excite | $\{0.1, 1, \mathbf{10}\}$ | $5e{-}5$ | $\{\mathbf{200}, 250\}$ | HORSE&BEAR |
| | CONFORM | $\{0.1, 1, \mathbf{10}\}$ | $5e{-}5$ | $\{\mathbf{200}, 250\}$ | HORSE&BEAR |
| | Divide&Bind | $\{\mathbf{0.1}, 1, 10\}$ | $5e{-}5$ | $\{\mathbf{200}, 250\}$ | HORSE&BEAR |
| | FOCUS | $\{0.01, 0.1, 1, 10, \mathbf{100}\}$ | $\{1e{-}4, \mathbf{5e{-}5}, 1e{-}5\}$ | $\{200, \mathbf{250}, 300\}$ | $\{\mathbf{HORSE\&BEAR}, \text{TWOOBJECT}\}$ |

### D.5 ADDITIONAL METRIC: OPEN-VOCABULARY DETECTION

As a complementary metric, we assess *subject presence* with OWL-V2 open-vocabulary detection (Minderer et al., 2023). For each prompt, we pass the subject strings as class queries and count an image as correct if *all* subjects are detected at least once. We report the fraction of images meeting this criterion.

Results for test-time control and fine-tuned models are shown in Tables 6 and 7. Both control algorithms increase subject presence over the base model. However, OWL-V2 is blind to attribute leakage and subject numerosity (it does not verify attributes or counts), so we exclude it from the main evaluation and report it only as a supportive metric here.

Table 6: OWL-V2 subject presence in percentage for test-time control.

| Heuristic | SD3.5 | FLUX |
|---|---|---|
| Base | 69.33% | 66.93% |
| Attend&Excite | 72.13% | 66.80% |
| CONFORM | **77.20**% | 67.87% |
| Divide&Bind | 70.80% | **68.53%** |
| FOCUS (Ours) | 74.27% | 68.27% |

Table 7: OWL-V2 subject presence in percentage for fine-tuned models.

| Heuristic | SD3.5 | FLUX |
|---|---|---|
| Base | 69.33% | 66.93% |
| Attend&Excite | **80.40%** | **74.93%** |
| CONFORM | 77.73% | 72.53% |
| Divide&Bind | 73.33% | 63.87% |
| FOCUS (Ours) | 78.53% | 74.66% |

# E  HUMAN STUDY

We evaluate whether metric gains translate to human preferences. Fifty participants each completed 40 prompt-conditioned, pairwise trials, resulting in 2,000 total judgments. In every trial, two images generated from the *same* prompt were shown side by side with the prompt; participants selected the image that better matched the prompt. The instruction shown was:

> Which image renders all subjects of the prompt correctly? If both do an equivalent good job, please pick the one you prefer visually.

To ensure sufficient rating density, we fixed the sampling seed to 0, yielding one image per method–prompt pair (pool of 150 prompts). Trials were balanced across backbone and setting: SD 3.5 vs. FLUX.1 and test-time control vs. fine-tuning each accounted for one quarter of the comparisons per participant. A screenshot of the interface is shown in Figure 8.

## E.1  ELO RATING COMPUTATION

We compute Elo ratings from the pairwise outcomes to obtain an across-method ranking, alongside win rates (fraction of pairwise wins). Elo is initialized at 1500 for all candidates and updated after each comparison with $K=32$. For a candidate $A$ with rating $R_A$ matched against $B$ with $R_B$, the expected score is

$$E_A = \frac{1}{1 + 10^{(R_B - R_A)/400}},\tag{37}$$

and the update is

$$R'_A = R_A + K(S_A - E_A)\tag{38}$$

where $S_A = 1$ for a win, $0$ for a loss, and $0.5$ for a draw. Higher Elo indicates stronger preference relative to alternatives. Win rate is reported as the proportion of head-to-head wins.

## F    EXTRA SAMPLES

### F.1    TEST-TIME CONTROL: STABLE DIFFUSION 3.5

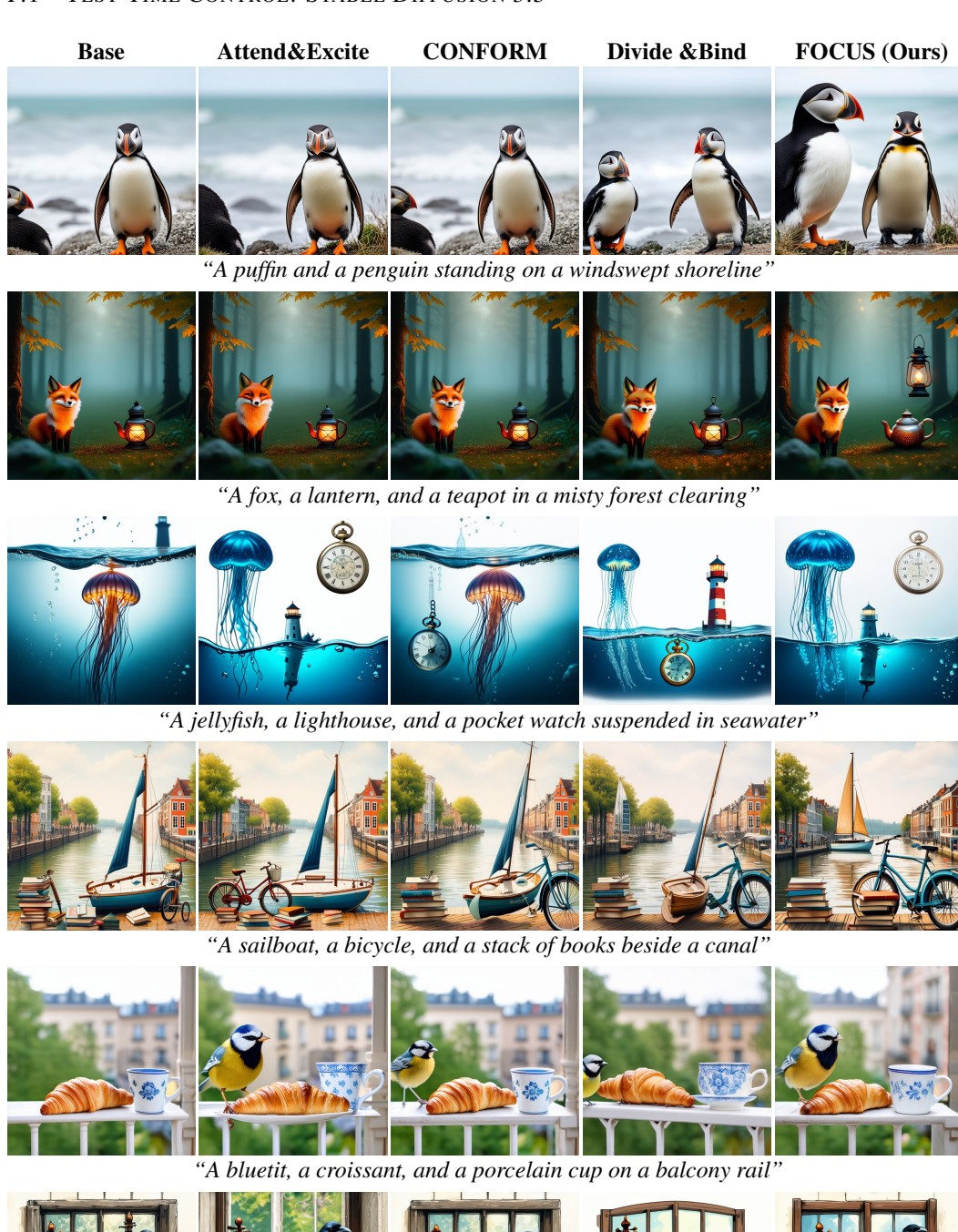

Figure 9: Stable Diffusion 3.5 samples with test-time control. All evaluated heuristics are shown at their optimal $\lambda$.

## F.2   TEST-TIME CONTROL: FLUX.1 [DEV]

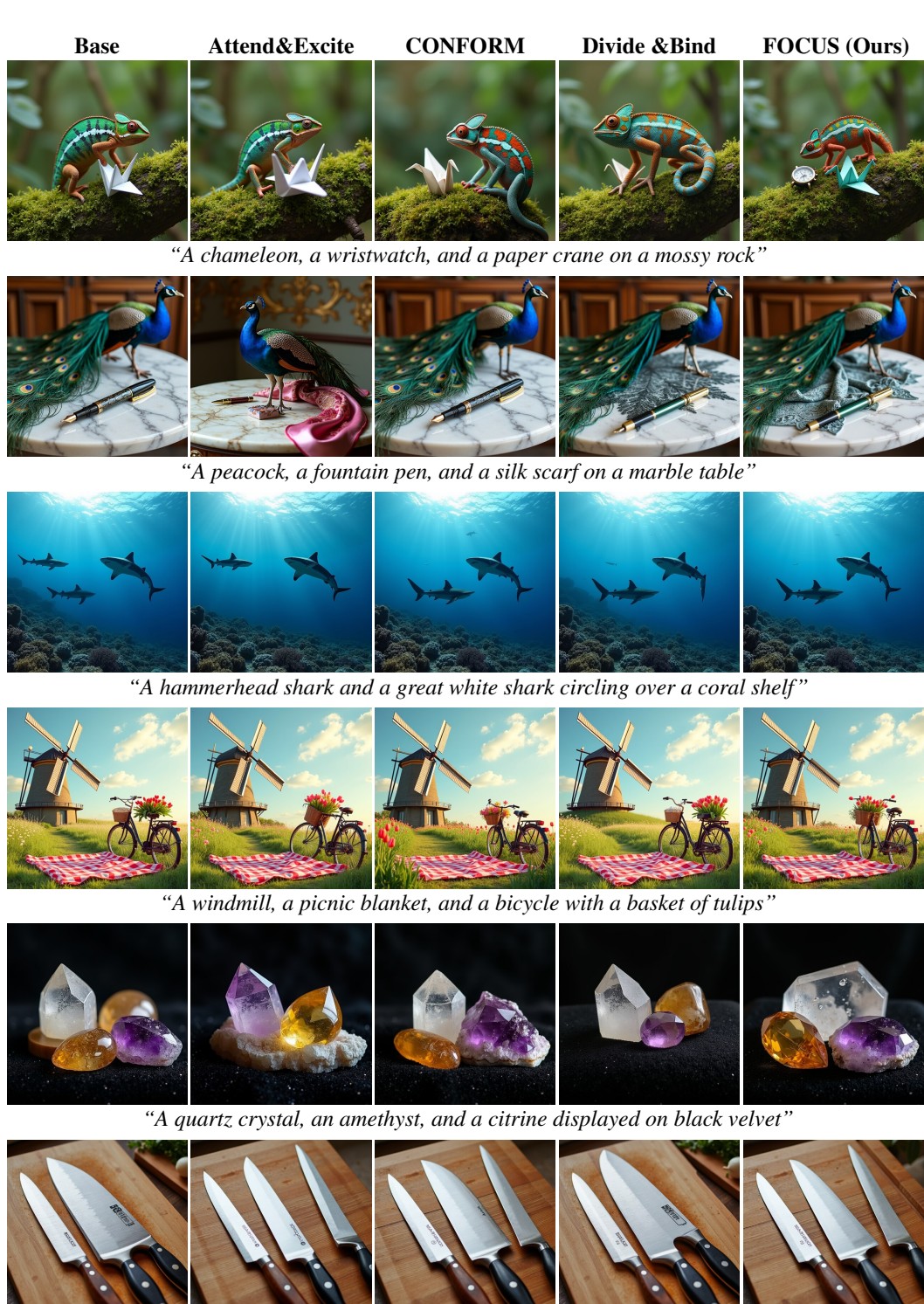

Figure 10: FLUX.1 dev samples with test-time control. All evaluated heuristics are shown at their optimal $\lambda$.

### F.3 FINE-TUNED: STABLE DIFFUSION 3.5

| Base | Attend&Excite | CONFORM | Divide &Bind | FOCUS (Ours) |
|------|---------------|---------|--------------|--------------|

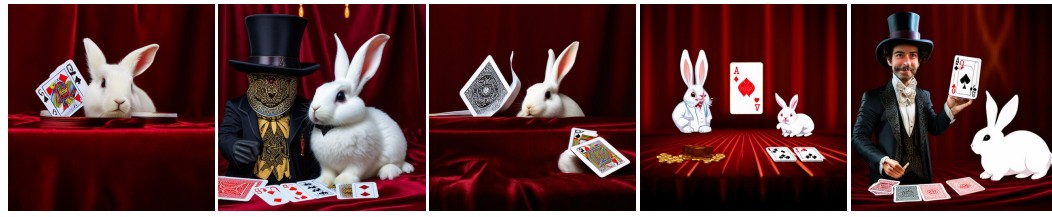

*"A Siberian Husky, an Alaskan Malamute, and a Samoyed trotting through fresh snow"*

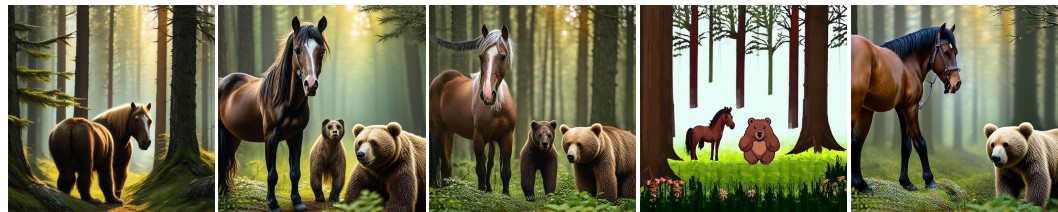

*"A magician, a white rabbit, and a deck of cards on a velvet stage"*

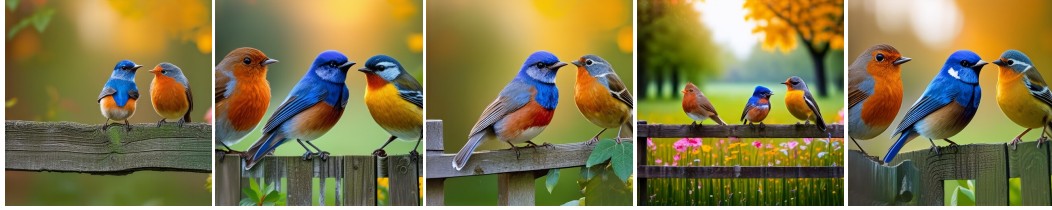

*"A horse and a bear in a forest"*

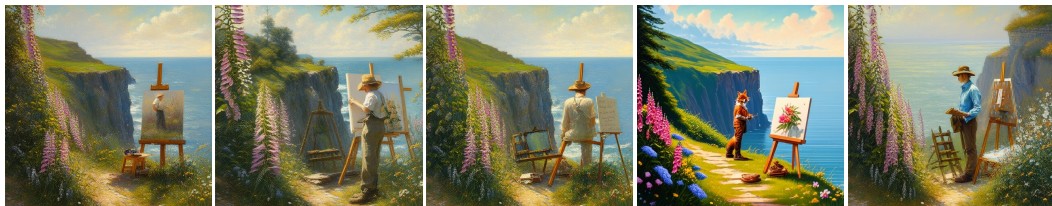

*"A robin, a bluebird, and a warbler perched on a garden fence at dawn"*

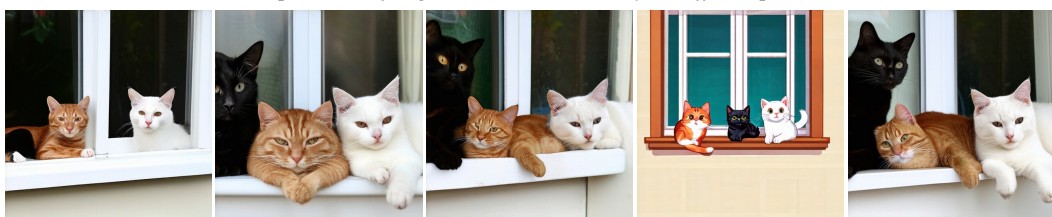

*"A painter, a foxglove, and an easel by a cliffside path"*

*"A black cat, an orange cat, and a white cat lounging on a windowsill"*

Figure 11: Sample results from Stable Diffusion 3.5 fine-tuned with each heuristic. Prompts were not seen during training to evaluate generalization. All images are generated with identical settings (seed, sampler, steps, guidance); each heuristic is shown at its optimal trained $\lambda$.

## F.4 FINE-TUNED: FLUX.1 [DEV]

Figure 12: Sample results from FLUX.1 [dev] fine-tuned with each heuristic. Prompts were not seen during training to evaluate generalization. All images are generated with identical settings (seed, sampler, steps, guidance); each heuristic is shown at its optimal trained $\lambda$.

