# OpenReview forum: "Optimal Control Meets Flow Matching: A Principled Route to Multi-Subject Fidelity"
_ICLR.cc/2026/Conference — ICLR 2026 Conference Withdrawn Submission_

### Official Review · Reviewer_HYS7 · 2025-10-28

**Soundness:** 2
**Presentation:** 2
**Contribution:** 2
**Rating:** 4
**Confidence:** 2

**Summary:**

This paper addresses multi-subject fidelity in text-to-image (T2I) generation by formulating the problem as stochastic optimal control (SOC) over flow matching (FM) dynamics. The authors observe that modern T2I models struggle with multi-subject prompts, exhibiting attribute leakage (attributes bleeding between subjects), identity entanglement (subjects merging), and subject omissions. To tackle these issues, they introduce a principled theoretical framework that views disentanglement as a control problem over trained FM samplers.

The paper presents two main algorithmic contributions: (1) a training-free test-time controller that perturbs the base velocity field with a single-pass gradient-based update derived from SOC optimality conditions, and (2) Adjoint Matching-based fine-tuning that regresses a lightweight control network (LoRA adapters) onto backward adjoint signals computed along frozen trajectories. Both methods balance proximity to the base model against a differentiable disentanglement objective. The authors introduce FOCUS (Flow Optimal Control for Unentangled Subjects), a probabilistic attention-based cost function using Jensen-Shannon divergence (JSD) to measure subject separation and within-subject coherence.

Empirically, the methods are evaluated on Stable Diffusion 3.5, FLUX.1, and Stable Diffusion XL using a curated 150-prompt dataset with 2-4 subjects per prompt. Results show consistent improvements in multi-subject alignment metrics (CLIP, SigLIP-2, BLIP, Qwen2-VL, PickScore, ImageReward) and human preference studies. Test-time control provides modest gains with minimal overhead (~2× inference time), while fine-tuning achieves stronger improvements, with models trained on as few as one prompt generalizing to unseen prompts. The framework unifies prior attention heuristics under a single optimization objective and extends to classical diffusion models via a flow-diffusion correspondence.

**Strengths:**

**Quality of Results:** Qualitative examples (Figures 1, 3, 4, 9-12) consistently show improved subject presence, reduced attribute leakage, and better separation compared to baselines. Quantitative results show consistent gains across metrics and human preferences, with FOCUS typically achieving best composite scores.

**Theoretical Foundation:** The paper's core strength is establishing the first principled, optimization-based framework for multi-subject disentanglement in T2I models.

**FOCUS Objective Design:** The probabilistic treatment of cross-attention as distributions is principled and well-motivated. Using Jensen-Shannon divergence respects the probabilistic structure of softmax attention, and the two-term formulation (within-subject coherence + between-subject separation) is intuitive. The bounded, normalized score enables meaningful comparison across prompts with different subject counts. The spatial smoothing and aggregation strategy (Section B) shows thoughtful engineering.

**Weaknesses:**

1. **FOCUS design choices lack justification**: While using JSD is principled, several decisions appear ad-hoc: (a) Why Gaussian smoothing specifically?  (b) Why average attention across blocks before scoring rather than score-then-average? (c) Why did the entropy regularizer fail (Figure 6 shows it pushes mass away from subjects—but why theoretically?)? (d) How sensitive is performance to these choices?

2.  **Incomplete diffusion model evaluation**: Claims of "extending to diffusion models via flow-diffusion correspondence" rest on one SDXL example (Figure 5). No quantitative evaluation, no comparison to diffusion-specific methods, no verification that theoretical correspondence translates to empirical effectiveness.

3. **Metric limitations**: Standard alignment metrics (CLIP, SigLIP) may not capture attribute leakage well (acknowledged for OWL-V2 but applies broadly). The composite score (Eq. 35) macro-averages relative improvements, which could mask failures on specific prompts or metrics. Human study uses only seed=0, potentially missing variation across seeds.

**Questions:**

1. **Approximation error analysis**: Can you provide theoretical or empirical bounds on the error introduced by the local adjoint approximation (Eq. 11)? How does this approximation degrade for longer time horizons or highly nonlinear dynamics?
2. **Alternative adjoint approximations**: Have you compared the left-Riemann approximation to other integration schemes (midpoint, trapezoid, higher-order) or to multi-step forward-backward passes? This would validate whether the simplest approximation suffices.
3. **FOCUS design ablations**:
* (a) What is the effect of removing Gaussian smoothing?
* (b) How does score-then-average compare to average-then-score for attention aggregation?
* (c) Why does the entropy regularizer push mass away from subjects (Figure 6)—can you explain this theoretically?
* (d) Is JSD specifically necessary, or would other divergences (KL, Wasserstein, Total Variation) work?

---

### Official Review · Reviewer_FE9W · 2025-10-30

**Soundness:** 2
**Presentation:** 3
**Contribution:** 3
**Rating:** 2
**Confidence:** 4

**Summary:**

This paper views flow matching from stochastic optimal control and introduces two architecture-agnostic algorithms to improve generative diffusion models, specifically for generation of multiple subjects. This paper proposes a parameter-efficient finetuning approach based on Adjoint Matching and a training-free method "FOCUS". FOCUS is a loss function that measures and minimizes subject entanglement. The experiments show advantages in improving attribute alignment and reducing object missing.

**Strengths:**

1. This work conducts experiments across different architectures, showing transferability.
2. The perspectives from stochastic optimal control is interesting and the proposed method by adapting adjoint matching and memoryless training is intuitive.

**Weaknesses:**

1. Most previous methods that the authors compare to do not natively design for SD3.5 and FLUX and are no longer SOTA. The authors should cite and compare to more recent and well-related papers and report quantitative comparisons with them, such as Self-Cross [1], EnMMDiT [2], TP-Blend [3].
2. This paper claims that it improves SDXL. However, there is no quantitative experiments supporting this. The only support comes from a single pair of qualitative results, which is insufficient.
3. The improvements over the previous methods are marginal, with usually worse BLIP text-to-text similarities. The authors introduce other metrics that other baselines do not report, such as PickScore and ImageReward, but do not explain the motivations. The authors should discuss why other metrics/benchmarks are flawed or unfair.

[1] Self-Cross Diffusion Guidance for Text-to-Image Synthesis of Similar Subjects, CVPR 2025

[2] Enhancing MMDiT-Based Text-to-Image Models for Similar Subject Generation, arXiv 2025

[3] TP-Blend: Textual-Prompt Attention Pairing for Precise Object-Style Blending in Diffusion Models, TMLR 2025

**Questions:**

1. The computational efficiency is not reported and the trade-off between computations and success rates is questionable. This method shows ~10% win rates over the backbone (SD3.5 or FLUX) but is at a cost of increasing computations. However, with test-time scaling by running the model multiple times until it successfully generates a certain number of samples, does the proposed method actually use fewer computations?

---

### Official Review · Reviewer_5xxm · 2025-11-04

**Soundness:** 3
**Presentation:** 3
**Contribution:** 3
**Rating:** 6
**Confidence:** 2

**Summary:**

This paper formulates the problem of multi-subject disentanglement as a stochastic optimal control (SOC) problem built upon a trained flow matching model. Leveraging the SOC theoretical framework, the authors propose two algorithms: (1) A training-free test-time controller that perturbs the velocity during sampling; (2) A fine-tuning approach using Adjoint Matching to regress the network to a backward adjoint signal. In addition, the authors design a metric called FOCUS to measure multi-subject entanglement. Experiments show that both algorithms improve the performance of text-to-image models (including SD3.5, FLUX, and SDXL) in multi-subject generation scenarios.

**Strengths:**

1. The method is theoretically grounded in the stochastic optimal control framework, offering a solid mathematical foundation.

2. The paper presents two complementary methods, one training-free and one fine-tuning based, providing users with a flexible trade-off between computational resources and inference efficiency.

**Weaknesses:**

1. Since FOCUS relies on cross-attention maps, the method appears to be limited to prompts that explicitly enumerate subjects (e.g., “a cat and a dog …”) and may not generalize well to prompts that specify quantities (e.g., “four cats …”).

2. As noted by the authors, the alignment metrics used in the paper may not fully capture attribute leakage issues. Existing benchmarks such as GenEval [1] and T2I-CompBench [2], which specifically target multi-subject generation, would provide a more comprehensive evaluation. It would be valuable to assess the proposed methods on these benchmarks to substantiate their effectiveness.

[1] Ghosh, Dhruba, Hannaneh Hajishirzi, and Ludwig Schmidt. "Geneval: An object-focused framework for evaluating text-to-image alignment." Advances in Neural Information Processing Systems 36 (2023): 52132-52152.

[2] Huang, Kaiyi, et al. "T2i-compbench: A comprehensive benchmark for open-world compositional text-to-image generation." Advances in Neural Information Processing Systems 36 (2023): 78723-78747.

**Questions:**

1. How does the proposed approach handle prompts that describe multiple subjects in terms of quantity, for example, “**four cats** sitting on a sofa”?

2. A minor typo: in Line 261, should the expression $P=\\{vp_1,\ldots,\boldsymbol p_n\\}$ be corrected to $P=\\{\boldsymbol p_1,\ldots,\boldsymbol p_n\\}$?

---

### Official Review · Reviewer_UHsk · 2025-11-06

**Soundness:** 1
**Presentation:** 2
**Contribution:** 3
**Rating:** 2
**Confidence:** 4

**Summary:**

This paper addresses multi-subject entanglement in text-to-image models. The authors introduce FOCUS, an intermediate time-step reward function based on the Jensen-Shannon divergence cost over attention maps, and propose two methods: (i) test-time control that adds gradient-based corrections to the velocity, and (ii) fine-tuning via Adjoint Matching that trains a control network. Experiments on SD 3.5, FLUX.1, and SDXL show improvements in multi-subject fidelity.

**Strengths:**

- **Well-motivated cost function.** FOCUS treats attention maps as probability distributions and uses Jensen-Shannon divergence to encourage within-subject coherence and between-subject separation, which is more principled than similarity metrics used in prior work.
- **Methodological contribution: Adjoint Matching for running cost optimization.** While Adjoint Matching (Domingo-Enrich et al., 2025) was developed for terminal costs, this paper adapts it to optimize intermediate running costs f(X_t, t) throughout the generation trajectory.
- Good empirical results including human user studies.

**Weaknesses:**

- **The test-time method appears to reduce to standard gradient-based guidance.** The core update (Eq. 17) is v* = v_base - (1-t)∇_X f(X_t, t), which appears mathematically equivalent to existing gradient-based guidance methods. Attend&Excite optimizes latents using ∇f, Divide&Bind uses gradients of coverage/binding costs, CONFORM uses contrastive gradients. In comparison, this paper
	- derives a weighted version of gradient-based guidance from SOC theory using control-theoretic notation (Hamiltonian, adjoint)
	- calls prior methods "heuristic" despite doing the same gradient-based updates
	- claims to provide "a principled route to adapt existing heuristics to modern FM models" but doesn't show whether this adaptation offers any advantage over simply applying original methods
- **No evidence the SOC formulation provides benefit beyond notational reframing.** To effectively validate this contribution, the following comparisons would be needed:
	- Rewards in SOC vs. Rewards in "standard guidance" formulation (shows if the added time-weighting term ($1-t$) adds value)
	- FOCUS vs. baseline costs in the same settings (ideally both SOC and standard guidance, show if FOCUS is generally the superior reward)
- **Contribution framing obscures the actual novelty.** The paper emphasizes the SOC derivation (Section 3.1-3.2), but this appears to be primarily mathematical repackaging of existing gradient guidance. The genuinely novel contributions are in my opinion (i) FOCUS as a cost function and (ii) Adjoint Matching for running cost optimization with generalization. These are somewhat buried. In my view, the paper should be repositioned around these contributions rather than claiming the control-theoretic framework as the primary novelty.
- **The fine-tuning setup is confusing and under-analyzed.** The paper trains on 1 prompt (HORSE&BEAR) or 15 prompts (TWOOBJECTS), evaluates on 150 diverse prompts, yet training on 1 prompt yields 85% better results than 15 prompts (Table 5). Shouldn't the fine-tuning generalize better when trained on more and more prompts? A larger analysis would make these experiments more convincing. Does performance improve with 50 or 100 prompts, or does it plateau/degrade? Without this, it's unclear whether minimal data is a feature or a limitation.

**Questions:**

- Can you provide direct comparisons between: (i) FOCUS in your SOC formulation vs. FOCUS in standard gradient-based guidance (e.g., Attend&Excite-style latent updates), and (ii) baseline methods in their original implementations vs. your SOC adaptations?
- Table 1 runs all baselines "at optimal λ", does this mean you reimplemented Attend&Excite, CONFORM, and Divide&Bind within the SOC framework or run "standard guidance"? What is their performance in original formulations on FM models?
- Beyond mathematical elegance, what does the SOC formulation provide that standard gradient-based guidance does not?
- Could you provide some intuition on fine-tuning on a single prompt outperforming fine-tuning on 15 prompts? This seems counter intuitive.
- Recently, noise-based test-time techniques have demonstrated strong performance on reward alignment benchmarks, particularly search-based [1] or optimization-based [2,3] methods outperforming standard reward guidance. Have you considered applying the FOCUS reward in one of these settings compared to standard guidance?

[1] Ma et al. "Inference-Time Scaling for Diffusion Models beyond Scaling Denoising Steps". CVPR 2025.

[2] Tang et al. "Inference-Time Alignment of Diffusion Models with Direct Noise Optimization". ICML 2025

[3] Eyring et al. "ReNO: Enhancing One-step Text-to-Image Models through Reward-based Noise Optimization". NeurIPS 2024.

---

### Note · Authors · 2025-11-14

I have read and agree with the venue's withdrawal policy on behalf of myself and my co-authors.